# Prevalence of Seasonal Influenza Vaccination in Chronic Obstructive Pulmonary Disease (COPD) Patients in the Balearic Islands (Spain) and Its Effect on COPD Exacerbations: A Population-Based Retrospective Cohort Study

**DOI:** 10.3390/ijerph17114027

**Published:** 2020-06-05

**Authors:** Laura Ruiz Azcona, Miguel Roman-Rodriguez, Montserrat Llort Bove, Job FM van Boven, Miguel Santibáñez Margüello

**Affiliations:** 1Global Health research group, University of Cantabria, 39005 Santander, Spain; laura.ruiz@unican.es (L.R.A.); miguel.santibanez@unican.es (M.S.M.); 2Primary Care Respiratory Research Group, Instituto de Investigación Sanitaria de las Islas Baleares (IdISBa), 07011 Palma, Spain; 3Balearic Primary Health Care Service (IBSalut), 07011 Palma, Spain; mllort@ibsalut.caib.es; 4Department of General Practice & Elderly Care Medicine, Groningen Research Institute for Asthma and COPD (GRIAC), University Medical Center Groningen, University of Groningen, 9713 GZ Groningen, The Netherlands; j.f.m.van.boven@rug.nl

**Keywords:** chronic obstructive pulmonary disease, influenza vaccination, COPD exacerbations, primary prevention

## Abstract

To determine the prevalence of influenza vaccination in chronic obstructive pulmonary disease (COPD) patients and its effect on COPD exacerbations, we conducted a retrospective population-based cohort study analyzing real-life data. We included all registered COPD patients ≥40 years old using respiratory medication during the study period (2012–2013). Influenza vaccination during the 2012/2013 campaign was the parameter studied. Moderate and severe exacerbations during 2013 were the dependent outcome variables. Logistic regression adjusting for age, gender, concomitant asthma diagnosis, COPD severity, smoking status, number of moderate and severe exacerbations the previous year, and comorbidities was performed, and 59.6% of the patients received seasonal influenza vaccination. The percentage of patients with exacerbations was higher among those vaccinated. Influenza vaccination had a statistically significantly negative (non-protective) crude effect favoring the risk of severe exacerbations: OR: 1.20 (95% CI; 1.05–1.37). This association diminished and lost statistical significance after adjustment: aOR: 0.93 (95% CI; 0.74–1.18). The protective effect in the analysis restricted to the epidemic period was not significant: aOR: 0.82 (95% CI; 0.58–1.16). We concluded that prevalence of influenza vaccination was suboptimal. In contrast with most of the available evidence, our results did not support a protective effect of influenza vaccination on the risk of admission for COPD exacerbation.

## 1. Introduction

Chronic obstructive pulmonary disease (COPD) prevalence is increasing over time. It is the fourth leading cause of death in the world and it will be the third by 2021. COPD exacerbations are episodes of temporary symptom worsening (e.g., breathlessness, cough, sputum production) that carry significant consequences for patients [1,2] Additionally, exacerbations are associated with an accelerated rate of lung function decline, reduced quality of life, and an increased mortality risk [3]. COPD exacerbations are heterogeneous and dependent on multiple risk factors [4]; however, generally, their frequency and severity increase as the disease worsens [5].

The fact that COPD exacerbations can be associated with influenza is of great public health interest. The most frequent causes of exacerbations are respiratory infections, most of them viral. In particular, influenza can be a causative virus for exacerbations, of which the more severe ones result in hospital admissions and even death [6,7]. Two systematic reviews recently analyzed the effectiveness of seasonal influenza vaccination in patients with COPD, concluding that the evidence supports a positive benefit–risk ratio for seasonal influenza vaccination in these patients, although most of the included trials were more than a decade old [8,9]. Considering the role of influenza in contributing to COPD exacerbations, the associated complications, and their related healthcare costs, immunization against influenza is recommended for all patients with COPD by major agencies and guidelines [1,10,11].

Despite influenza vaccination being recommended, vaccination rates are highly variable among different countries, including Spain [12,13,14,15,16,17], with further room for improvement [18]. On the one hand, those COPD patients who are likely to exacerbate could have more motivation to accept influenza vaccination [16,17]. On the other hand, vaccination rejection among COPD patients has been attributed to concerns about increased exacerbations or adverse reactions caused by the vaccine itself [13,16]. There are also many knowledge gaps regarding the impact of COPD severity and co-morbidity on influenza vaccine effectiveness. Therefore, additional studies are required [6,9].

Our primary aim is to determine the prevalence of influenza vaccination in COPD patients in a real-life population cohort in the Balearic Islands (Spain) and the association between history of influenza vaccination and COPD exacerbations.

## 2. Materials and Methods

Study design: Retrospective cohort study analyzing real-life data from the COPD population of the MAJOrca Real-world Investigation in COPD and Asthma cohort (MAJORICA-cohort). We reported our findings in line with Strengthening the Reporting of Observational Studies in Epidemiology (STROBE) guidelines for observational studies using routinely collected health data [19].

Data source: The MAJORICA-cohort contains data from all patients (≥18 years) with a primary care diagnosis of asthma and/or COPD in 2012 (*N* = 68,578), irrespective of health insurance, with at least two years follow-up available. Most recent follow-up data are from 2015. The cohort data are anonymously collected from the unique primary care system (eSIAP), all hospital systems, as well as the electronic prescription system (RELE) in the Balearic Islands, Spain [20,21,22].

Study population: Among the overall MAJORICA-cohort, patients who were using respiratory medication (ATC (Anatomical Therapeutic Chemical) code: R03) in 2012, 2013, and 2014 were firstly selected. From the included sample, all patients who had a physician diagnosis of COPD (ICD-9 codes: 491, 492, and/or 496) and were over 40 years of age were identified as patients with actively treated COPD.

Exposure: Patients were classified in two cohorts (exposed versus non-exposed) according to whether or not they had received influenza vaccination in the 2012–2013 campaign (between 18 October and 30 November 2012).

Outcome: The outcome of interest was COPD exacerbation. COPD exacerbation was defined as any episode involving an increase in patient’s baseline COPD symptoms (cough, pleghms, and/or dyspnea), requiring the prescription of an antibiotic and/or systemic corticosteroid (moderate exacerbation), or hospital admission for more than 24 h (severe exacerbation) [23,24]. The total frequency of exacerbations (moderate and severe) was quantified in 2012 and 2013. In a sensitivity analysis, exacerbations that occurred during the 2013 influenza epidemic seasonal period in our setting (7 January 2013 to 31 March 2013) [25] were also identified and quantified as a specific variable. In order to differentiate a new exacerbation from a previous treatment failure, each exacerbation had to be separated for at least 6 weeks from previous exacerbation initial treatment or 4 weeks from hospital discharge.

Co-variates: At baseline, for each patient, sociodemographic and clinical characteristics were collected, including sex, age, smoking habit, COPD obstruction severity (by GOLD grades 1–4), relevant comorbidities, respiratory treatments, and number and severity of COPD exacerbations in the previous year in order to identify the exacerbation “phenotype”. “Exacerbator phenotype” was defined as a patient who suffered from at least two moderate exacerbations or one severe exacerbation in a one-year period according to the main national and international guidelines definition [1,16,23,26]. “Non-exacerbator phenotype” patients were those having ≤1 moderate exacerbation and no severe exacerbations in a one-year period.

Statistical analyses: For categorical and discrete variables, proportions were estimated using Pearson’s Chi-squared tests for comparisons or Fisher’s exact tests when necessary. Quantitative variables were expressed as mean and standard deviation (SD) using the student’s *t*-test for comparisons after normal distribution was confirmed using the Shapiro–Wilk test. Dichotomous values (“at least one admission for COPD exacerbation”, “at least one moderate COPD exacerbation”, and “exacerbator phenotype or not” during the following year) were treated as dependent (effect) variables. To assess the crude univariate association between influenza vaccination and the risk of COPD exacerbations, crude odds ratios (OR) were estimated using unconditional logistic regression with 95% confidence intervals (95% CI). Non-vaccinated COPD patients were the reference category. Thus, an OR greater than one indicates that the vaccine is a risk factor for exacerbations. An OR less than one indicates that vaccination is protective, and an OR equal to one indicates a null effect on the risk of exacerbations the following year. To control for confounding bias, we predefined confounders related with the exposure (vaccination) and/or the outcome (exacerbations) in a stepwise regression analysis. The co-variables finally included were age (continuous variable), gender, concomitant asthma diagnosis, COPD severity (GOLD grades 1–4), smoking status (ordinal variable: non-smoker, former smoker, current smoker), number of moderate exacerbations in the previous year, number of severe exacerbations in the previous year, and the following comorbidities: heart failure, atrial fibrillation, cor pulmonale, anxiety, osteoporosis, allergic rhinitis, gastroesophageal reflux disease, and diabetes. Associations were stratified according to COPD severity based on forced expiratory volume in one second (FEV_1_%) predicted (GOLD grades 1–4) and restricted to patients with spirometry confirmed COPD (FEV_1_/FVC (Forced Vital Capacity) < 0.7) in a sensitivity analysis. We set the alpha error at 0.05, and all *p*-values were bilateral. All statistical analyses were performed using IBM SPSS Statistics, version 22.0 (IBM, New York, NY, USA).

Ethics: The study protocol PI17-07 was approved by the Primary Care Research Committee of Mallorca. All data came from real-life clinical files, but they were anonymized for analysis, making it impossible to identify individual patients.

## 3. Results

### 3.1. Population Characteristics

The flow chart for selecting study patients is shown in Figure 1 (R03 = respiratory medication).

In total, 12,396 patients fulfilled all study inclusion/exclusion criteria. Most patients were men (67.3%), overall mean age was 69.7 years, almost one third were current smokers, mean FEV_1_ (% predicted) was 62.7, and 24% were frequent exacerbators. Patients’ sociodemographic and clinical variables are shown in Table 1.

### 3.2. Prevalence and Characteristics of Influenza Vaccination

Overall, the influenza vaccination coverage in COPD patients was 59.6% (7393/12396). A comparison between vaccinated and non-vaccinated patients is presented in Table 1. The vaccination rate was slightly higher in men (62.6%) than women (53.6%), and vaccinated subjects were older than those non-vaccinated (71.9 versus 66.6 years, *p* < 0.001). There were also statistically significant differences in vaccination coverage as a function of BMI or smoking habit. Vaccination rates were slightly higher in more severe COPD patients (65.0% in GOLD 4) in comparison with milder obstructive ones (57.5% in GOLD 1). Vaccination coverage was also higher in patients with a history of moderate and/or severe exacerbations in 2012 (the same year of vaccination) compared to those with no exacerbations during the previous year. Suffering from each of the analyzed comorbidities was also associated with higher influenza vaccination coverage (Table 1). Main comorbidities in relation to influenza vaccination are described in Appendix A. 

### 3.3. Effects of Influenza Vaccination on COPD Exacerbations

Table 2 shows a descriptive analysis of the history of exacerbations (primary effect endpoint) during the following year as a function of influenza vaccination. The percentage of patients who suffered from moderate and/or severe exacerbations was higher among those vaccinated.

Influenza vaccination had a statistically significant negative (non-protective) crude effect favoring the risk of severe exacerbations: OR 1.20 (95% CI; 1.05–1.37). However, the association was towards the null effect (losing statistical significance) after adjusting for the main confounding variables: ORa 0.93 (95% CI; 0.74–1.18). After restricting to the 2013 influenza epidemic period, a non-significant protective effect against the risk of severe exacerbations was observed: ORa 0.82 (95% CI; 0.58–1.16). After stratifying according to GOLD obstruction severity (GOLD 1–4), a statistically significant negative (non-protective) adjusted effect seemed to be observed in the most severe patients, although statistical significance was not yielded when computing the 2013 influenza epidemic period only (Table 3).

Similar results were obtained restricting the analysis to patients with spirometry confirmed COPD (FEV_1_/FVC < 0.7), showing a null effect (non-negative, non-positive): ORa 0.97 (95% CI; 0.74–1.27) for the complete year, and ORa 0.81 (95% CI; 0.54–1.22) when the 2013 influenza epidemic period was analyzed (Appendix A).

A non-significant slightly negative (non-protective) association between influenza seasonal vaccination and COPD moderate exacerbations was found in both the 2013 complete year and in the influenza epidemic period (Table 4).

Similar results were shown in patients with spirometry confirmed COPD (FEV_1_/FVC < 0.7) (Appendix A). However, in very severe obstructive GOLD 4 patients, the non-protective effect was higher and statistically significant—ORa 4.54 (95% CI; 1.77–11.67) for the 2013 complete year—and an ORa 3.93 (95% CI; 1.39–11.13) was also obtained after computing the epidemic period only (Table 4), with similar results when restricting to spirometry confirmed COPD patients (Appendix A).

When the effect was computed as being “exacerbator phenotype or not”, that is, suffering from at least two moderate and/or severe exacerbations, an adjusted null effect of vaccination was observed in the overall population, and a negative (non-protective) effect was observed when restricting to very severe GOLD grade 4 patients (ORa 3.72 (95% CI; 1.48–9.33) (Table 5)) with similar results when restricting to spirometry confirmed COPD patients (Appendix A).

## 4. Discussion

Our results suggest that influenza vaccination in this concrete campaign (2012/2013) did not have a protective effect on the risk of severe and/or moderate exacerbations in the following year, not even if restricting the analysis to the epidemic influenza period only.

The absence of a preventive effect of seasonal influenza vaccination on moderate exacerbation risk is also supported by previous studies. However, in contrast to our results, no previous studies have shown an increased risk for severe exacerbations in very severe COPD patients after vaccination. In the recently published Spanish [16,17] or international studies [9,27], a protective effect on risk for severe exacerbations (admission for COPD exacerbation) or all-cause admissions was observed in those receiving seasonal vaccination. The higher protective effect being previously shown for the most severe COPD patients (GOLD 4) [17] is not supported by our results, given we found the opposite effect.

The influenza strains contributing to epidemics vary annually, and they influence the severity and the length of the epidemic seasonal period. Vaccination impact is supposed to be greater in seasons in which circulating strains match those used in vaccination schemes and lower in poorly matched seasons or so-called mismatched seasons. Influenza activity in Spain in the 2012–2013 season was moderate and was associated with a majority circulation of seasonal influenza B virus with a lower contribution of A(H1N1) virus throughout the pandemic wave [25]. The circulating viruses were consistent with the strains included in the 2012–2013 season vaccine for the northern hemisphere, except for the B viruses of the Victoria lineage not included in the vaccine recommended for the 2012–2013 season. However, the B viruses of the Victoria lineage circulated in minority, thus a lack of efficacy or mismatched vaccine would not explain the lack of preventive results in our study [25]. Since the 1996–1997 season, the influenza B virus has circulated predominantly in Spain in just two seasons, 2002–2003 and ten years later in 2012–2013. On the other hand, the 2012–2013 season had a late presentation compared to the previous ones, with an epidemic peak in mid-February. Although the epidemic wave was similar to the previous year, the 2012–2013 season was characterized by a prolonged period of intense flu virus circulation, with a percentage of flu-positive samples remaining above 50% for eleven consecutive weeks [25,28]. Regarding the immunogenicity of seasonal influenza vaccines in COPD patients, the seroconversion rates ranged from 34.4 to 61.3% for influenza B in published studies [29,30], being lower than the seroconversion rates for A/H1N1 (43 to 80.0%) or A/H3N2 (53.1 to 84.1%). Sero-protection rates for influenza B are also the lowest [9]. To what extent these peculiarities with respect to previous seasons could explain our results may deserve further attention.

The overall influenza vaccination coverage in the 2012–2013 campaign was 59.6%. This sub-optimal coverage is similar to those generally reported in developed countries, including Spain [12,13,17,31]. In our study, in line with other authors [16,17,32], we found that patients with several co-morbidities had greater vaccination coverage. It should also be mentioned that vaccination coverage among smokers was lower than among non-smokers, as also reported in other studies [16,17,33,34].

Vaccination rejection among COPD patients has been attributed to concerns about increased exacerbations or adverse reactions caused by the vaccine itself, which could explain the rejection, even among those with the more severe disease [13,16,34]. However, a greater incidence of exacerbation in the early weeks after vaccination was previously ruled out [35,36], and evidence for the safety of the vaccine seems to be conclusive [9].

Our study has several limitations.

Only a single influenza season (2012–2013) was studied. Looking at other seasons, especially those where recommended vaccine formulations more closely matched circulating virus strains, is needed to fully understand our observations. Similarly, data on serious influenza-related complications in the general population (vaccinated vs. unvaccinated) need to be taken into consideration.

Our study population was selected under the criterion that they should have used respiratory medication during the years of follow-up. This has advantages because it represents a population with active and relevant COPD. On the other hand, it makes it impossible, for example, to study mortality as a dependent variable in relation to influenza vaccination. A differential survival bias based on vaccination status could be a source of selection bias. If unvaccinated people had higher mortality and exacerbated more, the bias would underestimate the preventive effect of the vaccine. If, on the other hand, vaccinated patients, because they have more comorbidities or are older, die and exacerbate more, the bias would be in favor of the protective effect of the influenza vaccine.

The process for classifying record information into the outcome (i.e., moderate or severe exacerbations) was carried out by two physicians who were blinded to the influenza vaccination status of the study subjects. As the classification of the outcome was blinded to the exposed/non-exposed status of the study subjects, any misclassification of the outcome would have been non-differential towards the null association. It could be also a source of explanation for our results.

External validity is one of the main classical limitations of clinical trials, and this problem can also affect observational studies based on small samples [37,38,39]. A major strength of the present study is the inclusion of the entire population, that is, all patients diagnosed with COPD in the Balearic Islands, Spain, (+/− 1.1 million subjects) receiving primary health care and current treatment. This provides a useful COPD cohort representative for real-life care. It is improbable that bias arose through lack of blindness among patients’ care providers (they treated the patients blinded retrospectively before the development of the study). Another advantage is that we attempted to obtain the independent effect of flu vaccination in our epidemiological and statistical approach by controlling for potential confounding.

On the other hand, as with many registers and database studies, our data were collected from day-to-day clinical practice registers not specifically designed for the present study, and data collection was not under the researchers’ control. Therefore, some unmeasured variables could have confounded the results. Additionally, COPD misdiagnosis could promote a high risk of selection bias. To minimize selection bias, we performed a sensitivity analysis by restricting to only spirometry confirmed COPD cases, and we obtained similar results in this analysis.

## 5. Conclusions

The prevalence of influenza vaccination was suboptimal in the Balearic Islands (Spain). In contrast with the available evidence, our results did not support a protective effect of influenza vaccination on the risk of COPD exacerbations during the following year or even during the epidemic influenza season. A majority circulation of seasonal influenza B virus in the 2012–2013 season in Spain, with respect to previous seasons, could potentially explain our results and may deserve further attention. More research into the identification of specific groups where vaccination is effectively preventing exacerbations in COPD patients is needed.

## Figures and Tables

**Figure 1 ijerph-17-04027-f001:**
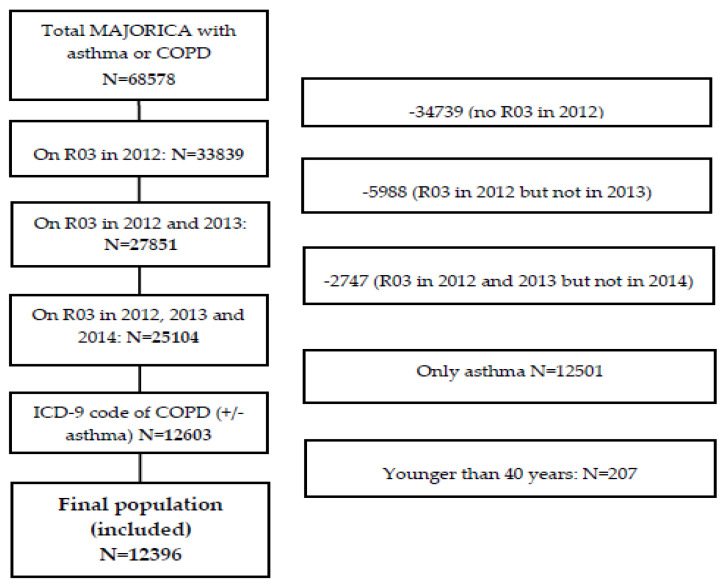
Flow chart for selection of study sample.

**Table 1 ijerph-17-04027-t001:** Main sociodemographic and clinical variables in relation to influenza vaccination during 2012/2013 seasonal campaign.

	Unvaccinated		Vaccinated		Total		
	*N* = 5003	% Row	*N* = 7393	% Row	*N* = 12,396	% column	*p* Value
**Sex**							
Women	1884	46.4%	2174	53.6%	4058	32.7%	<0.001
Men	3119	37.4%	5219	62.6%	8338	67.3%	
**Age**							
Mean [SD]	66.56	11.52	71.884	10.08	69.735	11	<0.001
**BMI**							
Mean [SD]	29.93	6.38	30.17	7.94	30.09	7.42	0.003
Normal weight (18.5–24.9)	594	41.1%	850	58.9%	1444	16.90% *	<0.001
Overweight (25–29.9)	1045	34.1%	2019	65.9%	3064	35.80%	
Obesity (≥30)	1381	34.8%	2583	65.2%	3964	46.30%	
Underweight (<18.5)	43	48.9%	45	51.1%	88	1.00%	
*Missing values*	*1940*	*50.6%*	*1896*	*49.4%*	*3836*	*30.9%*	
BMI > 21	2903	35.3%	5311	64.7%	8214	96.00%	<0.001
BMI ≤21	160	46.2%	186	53.8%	346	4.00%	
*Missing values*	*1940*	*50.6%*	*1896*	*49.4%*	*3836*	*30.9%*	
**Smoking habit**							
Never	1233	34.2%	2368	65.8%	3601	32.30% *	<0.001
Former	1439	32.9%	2930	67.1%	4369	39.20%	
Current	1735	54.6%	1443	45.4%	3178	28.50%	
*Missing values*	*596*	*47.8%*	*652*	*52.2%*	*1248*	*10.1%*	
**FEV_1_**							
Mean [SD]	64.19%	19.56	61.74%	19.79	62.66%	19.74	<0.001
**Obstruction severity (FEV_1_)**							
GOLD 1 (≥80%)	467	42.5%	633	57.53%	1100	19.60% *	<0.001
GOLD 2 (≥50–80%)	1133	37.8%	1862	62.2%	2995	53.40%	
GOLD 3 (≥30–49.9%)	449	33.7%	884	66.3%	1333	23.80%	
GOLD 4 (<30%)	64	35.0%	119	65.0%	183	3.30%	
*Missing values*	*2890*	*42.6%*	*3895*	*57.4%*	*6785*	*54.7%*	
**History of moderate exacerbations 2012**							
Number of exacerbations, mean [SD]	0.7	1	0.84	1.09	0.78	1.06	<0.001
None (0 exacerbations)	2800	43.0%	3714	57.0%	6514	52.5%	
1 exacerbation	1332	38.9%	2096	61.1%	3428	27.7%	
≥2 exacerbations	871	35.5%	1583	64.5%	2454	19.8%	
**Admitted for COPD exacerbation 2012**							
Number of COPD admissions, mean [SD]	0.08	0.34	0.1	0.39	0.09	0.37	0.003
None (0 COPD admissions)	4648	40.7%	6763	59.3%	11411	92.1%	0.01
1 COPD admission	292	36.5%	509	63.5%	801	6.5%	
≥2 severe exacerbations	63	34.2%	121	65.8%	184	1.5%	
**History of overall exacerbations 2012**							
Number of exacerbations, mean [SD]	0.79	1.15	0.94	1.26	0.88	1.22	<0.001
None (0 exacerbations)	2737	43.1%	3614	56.9%	6351	51.2%	<0.001
1 exacerbation	1254	38.6%	1993	61.4%	3247	26.2%	
≥2 exacerbations	1012	36.2%	1786	63.8%	2798	22.6%	
**Exacerbator phenotype 2012**							
No	3931	41.6%	5514	58.4%	9445	76.2%	<0.001
Yes	1072	36.3%	1879	63.7%	2951	23.8%	

* Valid percentage, excluding the missing values. BMI: body mass index, FEV_1_ forced expiratory volume in one second, COPD: chronic obstructive pulmonary disease. GOLD: Global Initiative for Obstructive Pulmonary Disease obstruction grades.

**Table 2 ijerph-17-04027-t002:** Influenza vaccination during the 2012/2013 campaign and exacerbations during the following year.

	Unvaccinated		Vaccinated		Total		
	*N* = 5003	%	*N* = 7393	%	*N* = 12,396	%	*p* Value
**Moderate COPD exacerbations 2013**							
Number of exacerbations.							
None (0 exacerbations)	2750	55.0%	3640	49.2%	6390	51.5%	<0.001
1 exacerbation	1279	25.6%	1943	26.3%	3222	26.0%	
≥2 exacerbations	974	19.5%	1810	24.5%	2784	22.5%	
**Severe COPD exacerbations 2013**							
Number of COPD admissions.							
None (0 COPD admissions)	4618	92.3%	6722	90.9%	11340	91.5%	0.003
1 COPD admission	320	6.4%	522	7.1%	842	6.8%	
≥2 COPD admissions	65	1.3%	149	2.0%	214	1.7%	
**Overall COPD exacerbations 2013**							
Number of exacerbations.							
None (0 exacerbations)	2701	54.0%	3564	48.2%	6265	50.5%	<0.001
1 exacerbation	1187	23.7%	1815	24.6%	3002	24.2%	
≥2 exacerbations	1115	22.3%	2014	27.2%	3129	25.2%	
**Exacerbator phenotype 2013**							
No	3842	76.8%	5311	71.8%	9153	73.8%	<0.001
Yes	1161	23.2%	2082	28.2%	3243	26.2%	

**Table 3 ijerph-17-04027-t003:** Crude and adjusted associations between “History of Influenza Vaccination during 2012–2013 campaign” and risk of “Admission due to COPD Exacerbation” in all patients with a diagnosis of COPD by obstruction severity during the complete year and the epidemic period.

	Severe Exacerbations (Admitted for COPD Exacerbation)											
	2013 Complete Year						2013 Epidemic Period Only					
	None	At Least One					None	At Least One				
	*N*= 11,340	*N* = 1056	OR	(95% CI)	ORa	(95% CI)	*N* = 11,997	*N* = 399	OR	(95% CI)	ORa	(95% CI)
***All COPD***												
***(N = 12,396)***												
*Unvaccinated*	4618	385	1	--	1	--	4846	157	1	--	1	--
*Vaccinated*	6722	671	1.20	(1.05–1.37)	0.93	(0.74–1.18)	7151	242	1.05	(0.85–1.28)	0.82	(0.58–1.16)
***GOLD 1 (N = 1100)***												
*Unvaccinated*	448	19	1	--	1	--	461	6	1	--	1	--
*Vaccinated*	603	30	1.17	(0.65–2.11)	0.87	(0.45–1.69)	623	10	1.23	(0.45–3.42)	0.94	(0.31–2.84)
***GOLD 2 (N = 2995)***												
*Unvaccinated*	1050	83	1	--	1	--	1095	38	1	--	1	--
*Vaccinated*	1741	121	0.88	(0.66–1.18)	0.72	(0.52–1.00)	1813	49	0.78	0.51–1.20	0.71	0.44–1.16)
***GOLD 3 (N = 1333)***												
*Unvaccinated*	398	51	1	--	1	--	428	21	1	--	1	--
*Vaccinated*	757	127	1.31	(0.93–1.85)	1.07	(0.71–1.61)	842	42	1.02	(0.59–1.74)	0.78	(0.42–1.44)
***GOLD 4 (N = 183)***												
*Unvaccinated*	55	9	1	--	1	--	60	4	1	--	1	--
*Vaccinated*	91	28	1.88	(0.83–4.28)	6.09	(1.77–20.97)	109	10	1.38	(0.41–4.58)	2.17	0.40–11.73

OR: crude odds ratio. ORa: odds ratio adjusted for age (continuous variable), gender, concomitant asthma diagnosis, smoking status (ordinal variable: non-smoker, former smoker, current smoker), number of moderate exacerbations the previous year, number of severe exacerbations the previous year, and the following comorbidities: heart failure, atrial fibrillation, cor pulmonale, anxiety disorder, osteoporosis, allergic rhinitis, gastroesophageal reflux disease, and diabetes. COPD obstruction severity: GOLD 1: mild; GOLD 2 moderate; GOLD 3: severe; GOLD 4 very severe.

**Table 4 ijerph-17-04027-t004:** Crude and adjusted associations between “History of Influenza Vaccination” and risk of moderate COPD Exacerbations” in all patients with a diagnosis of COPD by obstruction severity during the complete year and the epidemic period.

	Moderate Exacerbations											
	2013 Complete Year						2013 Epidemic Period Only					
	None	At Least One					None	At Least One			
	*N* = 6390	*N* = 6006	OR	(95% CI)	ORa	(95% CI)	*N* = 9345	*N* = 3045	OR	(95% CI)	ORa	(95% CI)
***Influenza Vaccination (N = 12,396)***												
*Unvaccinated*	2750	2253	1	--	1	--	3891	1112	1	--	1	--
*Vaccinated*	3640	3753	1.26	(1.17–1.35)	1.10	(0.97–1.24)	5460	1933	1.24	(1.14–1.35)	1.11	(0.96–1.28)
***GOLD 1 (N = 1100)***												
*Unvaccinated*	285	182	1	--	1	--	383	84	1	--	1	--
*Vaccinated*	329	304	1.45	(1.14–1.84)	1.21	(0.91–1.61)	475	158	1.52	(1.13–2.04)	1.33	(0.95–1.86)
***GOLD 2 (N = 2995)***												
*Unvaccinated*	648	485	1	--	1	--	892	241	1	--	1	--
*Vaccinated*	964	898	1.25	(1.07–1.44)	1.11	(0.93–1.31)	1404	458	1.21	(1.01–1.44)	1.05	(0.86–1.28)
***GOLD 3 (N = 1333)***												
*Unvaccinated*	203	246	1	--	1	--	323	126	1	--	1	--
*Vaccinated*	388	496	1.06	(0.84–1.33)	0.84	(0.65–1.09)	606	278	1.18	(0.92–1.51)	1.00	(0.75–1.33)
***GOLD 4 (N = 183)***												
*Unvaccinated*	37	27	1	--	1	--	47	17	1	--	1	--
*Vaccinated*	45	74	2.25	(1.21–4.19)	4.54	(1.77–11.67)	77	42	1.51	(0.77–2.95)	3.93	1.39–11.13

OR: crude odds ratio. ORa: odds ratio adjusted for age (continuous variable), gender, concomitant asthma diagnosis, smoking status (ordinal variable: non-smoker, former smoker, current smoker), number of moderate exacerbations the previous year, number of severe exacerbations the previous year, and the following comorbidities: heart failure, atrial fibrillation, cor pulmonale, anxiety disorder, osteoporosis, allergic rhinitis, gastroesophageal reflux disease, and diabetes. COPD obstruction severity: GOLD 1: mild; GOLD 2 moderate; GOLD 3: severe; GOLD 4 very severe.

**Table 5 ijerph-17-04027-t005:** Crude and adjusted associations between “History of Influenza Vaccination” and risk of being “Frequent exacerbator phenotype” the following year in all patients with a diagnosis of COPD by obstruction severity.

	2013 Frequent Exacerbator Phenotype					
	No	Yes				
	*N* = 9153	*N* = 3243	OR	(95% CI)	ORa	(95% CI)
Influenza Vaccination (2012–2013 Campaign)						
Total (*N*= 12,396)						
Unvaccinated	3842	1161	1	--	1	--
Vaccinated	5311	2082	1.30	(1.19–1.41)	1.02	(0.88–1.19)
GOLD 1 (*N* = 1100)						
Unvaccinated	384	83	1	--	1	--
Vaccinated	488	145	1.38	(1.02–1.86)	1.15	(0.80–1.63)
GOLD 2 (*N* = 2995)						
Unvaccinated	876	257	1	--	1	--
Vaccinated	1409	453	1.10	(0.92–1.31)	0.91	(0.74–1.12)
GOLD 3 (*N* = 1333)						
Unvaccinated	315	134	1	--	1	--
Vaccinated	569	315	1.30	(1.02–1.66)	1.03	(0.77–1.39)
GOLD 4 (*N* = 183)						
Unvaccinated	41	23	1	--	1	--
Vaccinated	65	54	1.48	(0.79–2.77)	3.72	(1.48–9.33)

OR: crude odds ratio. ORa: odds ratio adjusted for age (continuous variable), gender, concomitant asthma diagnosis, smoking status (ordinal variable: non-smoker, former smoker, current smoker), number of moderate exacerbations the previous year, number of severe exacerbations the previous year, and the following comorbidities: heart failure, atrial fibrillation, cor pulmonale, anxiety disorder, osteoporosis, allergic rhinitis, gastroesophageal reflux disease, and diabetes. COPD obstruction severity: GOLD 1: mild; GOLD 2 moderate; GOLD 3: severe; GOLD 4 very severe.

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
