# Peer review of "Prevalence of Seasonal Influenza Vaccination in Chronic Obstructive Pulmonary Disease (COPD) Patients in the Balearic Islands (Spain) and Its Effect on COPD Exacerbations: A Population-Based Retrospective Cohort Study"

_ijerph, 2020, doi:10.3390/ijerph17114027_

Round 1
Reviewer 1 Report
This is a carefully done if not too inspiring piece of epidemiological research coming from a large cohort of COPD patients from Balearic Islands (Spain). The value of this retrospective study is mostly contained in its unexpected observation of low-to-none benefit of vaccination against seasonal influenza for COPD patients. Authors have to commended for presenting results that go against conventional thinking. That said, this study has a number of limitations and some of them are pointed in the Discussion. Additionally, it needs to be said that authors looked only at a single influenza season of 2012-2013 and that looking at other seasons, especially when recommended vaccine formulation closely matched circulating virus strains may be needed to fully understand these observations. Similarly, data on serious influenza-related complications in general population (vaccinated vs. unvaccinated) need to be taken into consideration. English language, especially in the Abstract, Discussion and introductory part of the manuscript need to be improved as well, e. g., one would not say that influenza vaccination 'was the exposure of interest' (it was 'the parameter studied') or that 'vaccination impact is supposed to be greater in well matched seasons' ('seasons in which circulating strains matched those used in vaccination schemes'). A diligent read-through is therefore recommended with attention to missing or misplaced verbs or words, e.g. line 55: Despite influenza vaccination is strongly recommended - should be 'vaccination being recommended' or lines 249-250 'major strength of the present study is that our population study was the entire population that fulfilled inclusion criteria' - should be 'major strength of the present study is the inclusion of the entire population, etc.' This manuscript can be published after additional editing and expansion of Discussion section to incorporate the points mentioned above.
Author Response
This is a carefully done if not too inspiring piece of epidemiological research coming from a large cohort of COPD patients from Balearic Islands (Spain). The value of this retrospective study is mostly contained in its unexpected observation of low-to-none benefit of vaccination against seasonal influenza for COPD patients. Authors have to commended for presenting results that go against conventional thinking. That said, this study has a number of limitations and some of them are pointed in the Discussion. Additionally, it needs to be said that authors looked only at a single influenza season of 2012-2013 and that looking at other seasons, especially when recommended vaccine formulation closely matched circulating virus strains may be needed to fully understand these observations. Similarly, data on serious influenza-related complications in general population (vaccinated vs. unvaccinated) need to be taken into consideration. English language, especially in the Abstract, Discussion and introductory part of the manuscript need to be improved as well, e. g., one would not say that influenza vaccination 'was the exposure of interest' (it was 'the parameter studied') or that 'vaccination impact is supposed to be greater in well matched seasons' ('seasons in which circulating strains matched those used in vaccination schemes'). A diligent read-through is therefore recommended with attention to missing or misplaced verbs or words, e.g. line 55: Despite influenza vaccination is strongly recommended - should be 'vaccination being recommended' or lines 249-250 'major strength of the present study is that our population study was the entire population that fulfilled inclusion criteria' - should be 'major strength of the present study is the inclusion of the entire population, etc.' This manuscript can be published after additional editing and expansion of Discussion section to incorporate the points mentioned above.
Response:
We appreciate the opportunity to improve our manuscript according to the constructive and operational comments of the reviewer.
We have changed the sentence in the abstract to state that “Influenza vaccination during the 2012/2013 campaign was the parameter studied”.
We have included a new paragraph in the discussion section. We now state that only a single influenza season (2012-2013) is studied, and that looking at other seasons, especially when recommended vaccine formulation closely matched circulating virus strains is needed to fully understand our observations. Similarly, data on serious influenza-related complications in the general population (vaccinated vs. unvaccinated) need to be taken into consideration.
We have also changed the term ‘matched seasons' to 'seasons in which circulating strains matched those used in vaccination schemes' and we have modified also line 55 according to the rest of the suggestions…

Reviewer 2 Report
Azcona et al. studied the association between history of influenza vaccination and COPD exacerbations. The topic itself is certainly interesting and suit to International Journal of Environmental Research and Public Health. One major concern should be qualified before the publication:
This studied to compare incidence ratio of influenza vaccination among COPD population in Spain with referring to exacerbation events. Authors found influenza vaccination was significantly associated with COPD exacerbations. This may be interpreted as those who are likely to exacerbate conditions had more motivation to have flu shots. I can read in this way through results and discussion while am confuse to read introduction. It looks authors aimed to study the association between influenza and COPD exacerbation.Authors should amend to consistently state that the association between history of influenza vaccination and COPD exacerbation was studied.
Main reason to put negative remark is because study motivation is far from what authors presented in results. The study that COPD exacerbation is associated with influenza is of great interest in public health while an association between history of flu shot and COPD would be less. When authors can revise the manuscript to avoid this misconception, I'm happy to review again in spite of a little scientific benefit.
Author Response
Azcona et al. studied the association between history of influenza vaccination and COPD exacerbations. The topic itself is certainly interesting and suit to International Journal of Environmental Research and Public Health. One major concern should be qualified before the publication:
This studied to compare incidence ratio of influenza vaccination among COPD population in Spain with referring to exacerbation events. Authors found influenza vaccination was significantly associated with COPD exacerbations. This may be interpreted as those who are likely to exacerbate conditions had more motivation to have flu shots. I can read in this way through results and discussion while am confuse to read introduction. It looks authors aimed to study the association between influenza and COPD exacerbation. Authors should amend to consistently state that the association between history of influenza vaccination and COPD exacerbation was studied.
Main reason to put negative remark is because study motivation is far from what authors presented in results. The study that COPD exacerbation is associated with influenza is of great interest in public health while an association between history of flu shot and COPD would be less. When authors can revise the manuscript to avoid this misconception. I'm happy to review again in spite of a little scientific benefit.
Response:
We agree and appreciate the reviewer's comment.
We have added a remark in the introduction section stating that “The fact that COPD exacerbations can be associated with influenza is of great public health interest”. We have also added a paragraph to explain that those who are likely to exacerbate could have had more motivation to have flu shots. Lastly, we have changed the last sentence in our objective to consistently state that the association between history of influenza vaccination and COPD exacerbation was studied.

Round 2
Reviewer 2 Report
Thank you for inviting me to re-review of the manuscript. That work has improved significantly and I believe that is scientifically fair to present. I don't have further comments on.